# Visuo-Perceptual and Decision-Making Contributions to Visual Hallucinations in Mild Cognitive Impairment in Lewy Body Disease: Insights from a Drift Diffusion Analysis

**DOI:** 10.3390/brainsci10080540

**Published:** 2020-08-11

**Authors:** Lauren Revie, Calum A Hamilton, Joanna Ciafone, Paul C Donaghy, Alan Thomas, Claudia Metzler-Baddeley

**Affiliations:** 1Cardiff University Brain Research Imaging Centre (CUBRIC), School of Psychology, Cardiff University, Maindy Road, Cardiff CF24 4HQ, UK; metzler-baddeleyc@cardiff.ac.uk; 2Translational and Clinical Research Institute, Newcastle University Biomedical Research Building, Campus for Ageing and Vitality, Newcastle Upon Tyne NE4 5PL, UK; c.hamilton3@newcastle.ac.uk (C.A.H.); joanna.ciafone@newcastle.ac.uk (J.C.); paul.donaghy@newcastle.ac.uk (P.C.D.); alan.thomas@newcastle.ac.uk (A.T.)

**Keywords:** Lewy body disease, mild cognitive impairment, attention, visual hallucinations, perception, drift diffusion, vision

## Abstract

*Background:* Visual hallucinations (VH) are a common symptom in dementia with Lewy bodies (DLB); however, their cognitive underpinnings remain unclear. Hallucinations have been related to cognitive slowing in DLB and may arise due to impaired sensory input, dysregulation in top-down influences over perception, or an imbalance between the two, resulting in false visual inferences. *Methods:* Here we employed a drift diffusion model yielding estimates of perceptual encoding time, decision threshold, and drift rate of evidence accumulation to (i) investigate the nature of DLB-related slowing of responses and (ii) their relationship to visuospatial performance and visual hallucinations. The EZ drift diffusion model was fitted to mean reaction time (RT), accuracy and RT variance from two-choice reaction time (CRT) tasks and data were compared between groups of mild cognitive impairment (MCI-LB) LB patients (*n* = 49) and healthy older adults (*n* = 25). *Results:* No difference was detected in drift rate between patients and controls, but MCI-LB patients showed slower non-decision times and boundary separation values than control participants. Furthermore, non-decision time was negatively correlated with visuospatial performance in MCI-LB, and score on visual hallucinations inventory. However, only boundary separation was related to clinical incidence of visual hallucinations. *Conclusions:* These results suggest that a primary impairment in perceptual encoding may contribute to the visuospatial performance, however a more cautious response strategy may be related to visual hallucinations in Lewy body disease. Interestingly, MCI-LB patients showed no impairment in information processing ability, suggesting that, when perceptual encoding was successful, patients were able to normally process information, potentially explaining the variability of hallucination incidence.

## 1. Introduction

Dementia with Lewy bodies (DLB) is clinically characterized by four core clinical features, frequent cognitive fluctuations, complex visual hallucinations, rapid eye movement sleep behavior disorder (RBD), and Parkinsonism [1]. Cognitively, DLB patients also experience disproportionate impairment in attention and visual perceptual abilities in comparison to both older adults and patients with Alzheimer’s disease (AD) [1,2,3,4]. The question arises how these cognitive and perceptual impairments relate to clinical symptoms in DLB. For instance, visual hallucinations may arise from bottom up visuoperceptual and top-down attentional impairments [5], and similarly, cognitive fluctuations may be related to attentional deficits [6].

Mild cognitive impairment with Lewy bodies (MCI-LB) refers to individuals experiencing mild cognitive impairment which is likely to progress to DLB [7]. MCI refers to the stage between normal aging and dementia, in which cognitive decline occurs, but activities of daily living are relatively preserved [8]. Patients may be diagnosed with possible MCI-LB if one core clinical feature or biomarker is present, or patients may be diagnosed with probable MCI-LB if two or more clinical features are present, or one clinical feature plus one biomarker according to the proposed MCI-LB criteria [7]. Cognitively, MCI-LB patients experience marked impairments in processing speed, executive dysfunction and visuospatial functions [3]. A greater slowing of response times (RT) in DLB compared with AD patients and healthy older controls, particularly in tasks with increasing complexity of cognitive processing has been frequently reported (i.e., in choice reaction task; CRT) [9,10]. For instance, DLB patients show disproportional slowing in flanker performance, but can exhibit the flanker effect when alerted to the trial [11]. Moreover, DLB patients exhibit large intraindividual fluctuation in RT performance, that may underpin cognitive fluctuation [10,12]. However, not all studies have found variability in CRT performance to be related to visual or attentional performance, thus this link remains unclear [13]. Currently, the nature of RT slowing and variability in DLB and their relationship to clinical symptoms is not well understood. To address this, the present study adopted the diffusion drift model (DDM) to gain a better understanding of the nature of RT slowing in DLB patients.

Based on the assumptions that decisions occur over time and are prone to errors, the DDM can be applied to CRTs. The drift diffusion model [14,15] is a sequential-sampling model, which assumes that information driving a decision is accumulated over time, until it reaches one of two response boundaries (denoted by “upper” and “lower” response boundaries in Figure 1) and forms the ultimate response (for example, a “left” or “right” response would be the upper or lower response boundaries in a model as depicted in Figure 1). According to the diffusion model, overall processing is segmented into several components that contribute to ultimate performance: (1) first, components of processing that do not include active decision-making, i.e., such as perceptual encoding and response execution which are represented by the “non-decision time (t)”, (2) second, a criterion threshold that information accumulates to, which is represented by the “boundary separation (a)”, and finally, (3) the rate that information accumulates towards a decision, estimated by the “drift rate (v)”. The DDM assumes that the accumulation of information (drift rate) during a decision process is not constant but varies over time, represented by (s). This variability also contributes to error responses, where the accumulation of information can reach the incorrect boundary. For example, in the context of a flanker task in which stimuli are either congruent and thus “easy”, or incongruent and thus “harder”, drift rates will tend to be more prone to variation in harder conditions, and therefore response times are slower and have a higher probability of reaching an incorrect response boundary.

The DDM has been extensively applied in aging research due to the effect of aging on the speed accuracy trade off. Most consistent findings from drift diffusion modelling in older adults show increased boundary separation values and lengthening of non-decision times despite no difference in accuracy between older and younger adults [17,18]. Drift rate remains relatively preserved in older adults, indicating a shift in response strategy to a more cautious decision-making approach.

The DDM was applied to CRT performance of Parkinson’s (PD) patients with hallucinations relative to PD without hallucinations and healthy controls. Slower drift rates were found in PD patients with hallucinations, and shorter perceptual encoding times were found in all PD patients compared to controls, suggesting that the accumulation of evidence was hindered by perceptual encoding problems [19]. The authors proposed that impaired sensory evidence accumulation may lead to reduced information processing quality and an over-reliance on top down processing in PD, which in turn may underpin visual hallucinations.

The question arises whether DLB patients share similar impairments in perceptual and evidence accumulation as PD patients and whether these can be detected at the MCI-LB stage. By employing a variation of the DDM—the EZ model [20] to assess elements of CRT that may indicate a primary perceptual or attentional impairment, we may be able to narrow down those cognitive processes that underpin visual impairments, cognitive slowing and visual hallucinations in MCI-LB patients. This is of importance, as hallucinations in Lewy body disorders have been proposed to be the product of impaired sensory input and dysfunctional attentional in terms of faulty top-down attentional control [21,22], but the relative contribution of these processes remains unclear. Moreover, visual hallucinations are a predictor of cognitive decline in PD and have a significant effect on the quality of life and treatment of patients and caregivers in both PD and DLB [23], therefore there is a clear clinical need in understanding this symptom. As both DLB and PD patients often experience visual hallucinations, and share pathological features, it was hypothesized that MCI-LB patients would show similar impairments in drift rate and perceptual encoding times as those previously reported for PD patients with hallucinations.

## 2. Materials and Methods

### 2.1. Participants

Patients were part of the Lewy-Pro cohort [24]. Participants have been described in detail previously [24,25]. Briefly, individuals were recruited from memory clinics in the North East of England and Cumbria, were over 60 years of age and met NIA-AA criteria for MCI. All participants were clinically assessed for core features of DLB by an experienced psychiatrist (PD). Additional assessments included neuropsychological assessment and dopaminergic imaging using 123I-N-fluoropropyl-2β-carbomethoxy-3β-(4-iodophenyl) single-photon emission computed tomography (FP-CIT SPECT), which were rated normal/abnormal blind to clinical information by an expert panel.

Diagnosis was made by a consensus panel of old age psychiatrists, and participants were categorized into one of two groups; probable MCI-LB in which patients had two or more of the core diagnostic features of DLB or one feature and an abnormal FP-CIT scan, and possible MCI-DLB in which patients had one core feature or an abnormal FP-CIT scan. These diagnoses match the categories of probable MCI-LB and possible MCI-LB in the recently published international consensus criteria [7]. Patients were excluded if they had dementia, an MMSE score <20, a CDR score of >0.5, Parkinsonism which developed more than one year prior to cognitive impairment, or evidence of stroke, neurological or medical conditions which would affect performance in assessments. For this investigation, data from a subset of 50 participants described previously by Donaghy et al., [24] were accessed, including CRT data, and clinical assessments. One participant (probable MCI-LB) was excluded due to limited CRT data.

Healthy controls were recruited from the School of Psychology volunteer’s panel at Cardiff University, local age interest groups and the Join Dementia Research community in Cardiff, South Wales, UK. Controls had no visual disturbances, no history of psychiatric illness, no current diagnosis of dementia or cognitive impairment, and were over the age of 60 years. A group of younger control participants was also included in the analysis, which allowed us to contrast disease and normal aging-related differences in the drift diffusion parameters. These participants were recruited from a School of Psychology undergraduate student database at Cardiff University, and were subject to the same exclusion criteria as the older control participants. Table 1 shows demographic information of the cohort.

### 2.2. Materials

Lewy-Pro participants’ general cognitive performance was assessed with the Addenbrooke’s Cognitive Exam (ACE-R; [26], verbal fluency with the FAS Verbal Fluency and Graded Naming Tests [27], attention-switching with the Trail-making test parts A and B and verbal memory with the Rey Auditory Verbal learning test (RAVLT) [28]. Participants completed computerized assessments of CRT, digit vigilance and visuoperceptual functions, in addition to clinical assessments including the Geriatric Depression Scale (GDS) [29], Clinician Assessment of Fluctuations [30], Dementia Cognitive Fluctuations Scale (DCFS) [31], Neuropsychiatric Inventory (NPI) [32], North East Visual Hallucinations Interview (NEVHI) [33], Unified Parkinson’s Disease Rating Scale (UPDRS) [34], and the Instrumental Activities of Daily Living scale (ADL) [35]. Control participants were also assessed using a computerized assessment of choice reaction time, the ACE-R, CAF and NEVHI.

### 2.3. Procedure

MCI-LB participants completed a simple CRT ‘left or right’ arrow response task (30 trials) as described in Donaghy et al. [24]. RT data for each control participant were collected using a CRT flanker task [36]. The task consisted of five horizontal arrows presented on the screen. Participants were instructed to attend to the central arrow and respond to the direction the target arrow pointed to, using either left or right arrow keys. Central arrows were flanked either by lines (neutral condition) or by arrows that either pointed in the same direction as the central arrow (congruent condition) or in its opposite direction (incongruent condition). Targets were presented on the screen for 1700 ms, and participants were asked to respond as quickly and as accurately as possible. Participants completed 96 trials in each block, for a duration of 5 blocks, with a rest of 30 s between each block, totaling 480 trials. For the purposes of the DDM analyses and to account for differences in complexity of tasks between groups, EZ model parameters for the control participants were calculated only from the RT data of the “neutral” condition in the CRT flanker task (number trials =160).

The means of RT, accuracy and RT variance were generated for each participant across all completed trials. Participants were excluded from the analysis if they did not complete the task (*n* = 1). A simplified version of the original drift diffusion model, the EZ DDM [20] was then used to estimate each participants’ drift rate [v], boundary separation [a], and non-decision time [t] from mean and variance of RTs in correct decision trials and from accuracy (proportion correct). The EZ model outperforms other DDM in fitting CRT data from limited trial numbers (<100), and thus is particularly suitable for the modelling of data from patients with reduced cognitive abilities [37]. The EZ model has also been shown to be sensitive to responses which may arise from sources outside the diffusion process assumptions [38], although also see Ratcliff et al. for further information [39].

### 2.4. Statistical Analyses

Group differences in mean RT, accuracy, RT variance, and the three EZ DDM parameters (non-decision time, boundary separation, and drift rate) were tested between young and older participants to assess age-related effects, and between older participants and possible MCI-LB and probable MCI-LB groups respectively, to assess disease-related effects. Variance was assessed using Levene’s test for homogeneity of variance. Due to the high variance in patient data, non-parametric Kruskall–Wallis tests, with post-hoc Mann–Whitney U tests were employed for all group comparisons. All multiplicity of post-hoc testing was corrected with 5% False Discover Rate (FDR) using the Benjamini–Hochberg procedure [40] where *p_cor_* is reported as FDR corrected *p*-values.

Following this, linear hierarchical stepwise regression models were fit to demographic and DDM data in order to determine the best predictors of clinical measures in both possible and probable MCI-LB groups. Spearman rho correlations were calculated between significant predictors and clinical measures to assess the direction of effects.

Patient groups were also categorized on the basis of their NEVHI score, with a high NEVHI indicating greater incidence of visual disturbances. This allowed us to compare drift parameters between those patients who experienced symptoms without visual hallucinations, (NEVHI score = 0; NEVHI−, *n* = 29) and those who experienced symptoms with visual hallucinations (NEVHI score > 1; NEVHI+, *n* = 20), see [19]. In addition, these comparisons were also conducted between patients with clinically rated absence (*n* = 37) or presence (*n* = 12) of complex visual hallucinations. Mann–Whitney U tests were used to assess group differences in DDM parameters.

## 3. Results

### 3.1. Demographics

Kruskal–Wallis H tests revealed group differences in age (χ2 (2) = 12.558, *p* = 0.002) and education (χ2 (2) = 22.280, <0.001). Both probable and possible MCI-LB groups were significantly older than healthy control participants (probable: U = 189, *p_cor_* < 0.001; possible: U = 65, *p_cor_* = 0.02). In addition, control participants had significantly more years of education than possible (U = 241.5, *p_cor_* < 0.001) and probable MCI-LB groups (U = 661.5, *p_cor_* = 0.001).

### 3.2. Mean RT, Accuracy and Variance Differences between Older Versys Younger Controls and between Older Controls and MCI-LB Groups

Kruskal–Wallis H tests showed group differences in mean RT (χ2 (2) = 15.747, *p* < 0.001), mean accuracy (χ2 (2) = 16.927, *p* < 0.001), mean RT variance (χ2 (2) = 7.589, *p* = 0.022) and standard deviation in RT (χ2 (2) = 26.503, *p* < 0.001). Younger participants had significantly lower RTs than older controls (U = 499, *p_cor_* < 0.001) (Figure 2A); however, younger and older adults did not significantly differ in accuracy (*p* = 0.17) (Figure 2B). Both RT variance (*p* = 0.25) and RT standard deviation (*p* = 0.621) did not differ between younger and older adults.

Mean RTs were significantly lower in the older control relative to the probable MCI-LB group (U = 157, *p_cor_* < 0.001) (Figure 2A). Healthy older controls also had higher accuracy scores than possible (U = 46, *p_cor_* < 0.001) and probable MCI-LB patients (U = 215, *p*_cor_ < 0.001) (Figure 2B). Reaction time variance did not differ between healthy older controls and possible MCI-LB (*p_cor_* = 0.82), or probable MCI-LB (*p_cor_* = 0.64). Standard deviation was significantly greater in possible MCI-LB (*p_cor_* = 0.0015) and probable MCI-LB (*p_cor_* < 0.001) in comparison to healthy older controls.

### 3.3. Group Differences in DDM Parameters

Kruskal–Wallis H tests showed significant differences between all groups in non-decision time (χ2 (2) = 20.429, *p* < 0.001) and boundary separation values (χ2 (2) = 9.808, *p* = 0.007), but not drift rates (χ2 (2) = 0.174, *p* = 0.917). Post-hoc Mann–Whitney U tests revealed that older adults had significantly longer boundary separation values than younger adults (U = 415, *p_cor_* = 0.04). No significant difference in non-decision time (*p* = 0.11) between older and younger adults were observed.

Post-hoc Mann–Whitney U tests showed significantly lower boundary separation values in both possible MCI-LB (U = 53, *p_cor_* = 0.004), and probable MCI-LB (U = 240, *p_cor_* = 0.009) in comparison to older adults. Moreover, greater non-decision times were observed in possible MCI-LB (U = 56, *p_cor_* = 0.005), and probable MCI-LB (U = 125, *p_cor_* ≤ 0 0.001) in comparison to older adults. In addition, no significant differences in boundary separation value (*p_cor_* = 0.61) and non-decision time (*p_cor_* = 0.40) were observed between possible and probable MCI-LB. Mann–Whitney U tests also showed significant differences in variances between drift rate, non-decision time and boundary separation values between older adults, possible MCI-LB and probable MCI-LB groups (drift: U = 23.5, *p_cor_* < 0.001; non-decision: U = 444, *p_cor_* ≤ 0.001; boundary: U = 420, *p_cor_* < 0.001). Additionally, Levene’s test of variance showed significantly greater variation in non-decision time (F (2, 69) = 8.41, *p* < 0.001), boundary separation (F (2, 68) = 3.26, *p* = 0.041), and drift rate (F (2, 71) = 9.809, *p* < 0.001) between control participants and the probable MCI-LB group (Figure 3).

### 3.4. Effect of Age and Education on Group Differences

The potential influence of age and education on group differences was assessed given that the participants with possible and probable MCI-LB were older and less educated compared to the older control group. This was done by repeating the patient-control comparisons for a subset of control participants aged over 70 (*n* = 11), in which age (χ2 (2) = 1.387, *p* = 0.50) and education (χ2 (2) = 5.980, *p* = 0.06) did not significantly differ from the patient groups. The same pattern of results was observed for these sub-group comparisons, in which mean RT was significantly lower (χ2 (2) = 11.059, *p* = 0.004) and mean accuracy significantly higher in older adults (χ2 (2) = 12.139, *p* = 0.002) than both patient groups. Similarly, mean non-decision time was significantly lower in older adults (χ2 (2) = 12.844, *p* = 0.002) and boundary separation greater in older adults (χ2 (2) = 6.253, *p* = 0.044), with no significant difference in drift rate (χ2 (2) = 0.082, *p* = 0.960). Thus, age and education did not appear to significantly affect the differences between patients and older control participants.

### 3.5. DDM Parameters Are Predicted by Clinical Assessments of Visual Perception

Data from both possible and probable MCI-LB patients were entered into a series of hierarchical stepwise linear regressions. First, demographic information including education, gender, age, and MCI-LB group (probable, possible) were entered as predictors of either ACE subscales, NEVHI, CAF, or DCFS scores. Following this, DDM parameters were added to the models as predictors.

A diagnosis of probable MCI-LB (adj R^2^ = 0.12, beta = −0.30, *p* = 0.035) along with years of education (adj R^2^ = 0.05, beta = 0.31, *p* = 0.028) were significant predictors for the ACE visuospatial score. The addition of DDM parameters to the model revealed that non-decision time (F (3, 45) = 5.413, R^2^ = 0.265, adj R^2^ = 0.129, beta = −0.333, *p* = 0.014) significantly predicted ACE visuospatial performance in the patient groups. The addition of non-decision time as a predictor in the model was also significant (Delta R^2^ = 0.148, F change (1, 47) = 8.136, *p* = 0.006).

Moreover, male gender (adj R^2^ = 0.06, beta = −0.295, =0.036) to significantly predicted NEVHI score. The addition of DDM parameters to the model showed that non-decision time approached significance but did not survive corrections for multiple comparisons (adj R^2^ = 0.062, beta = 0.249, *p* = 0.06). Non-decision time was returned as a predictor of DCFS score but did not reach statistical significance (*p* = 0.072). Moreover, drift rate was returned as a predictor of CAF score, but did not reach statistical significance (*p* = 0.321).

Spearman’s Rho correlational analysis showed that non-decision time was negatively correlated with ACE Visuo-spatial subscale (rho = −0.38, *p* < 0.001), but drift rate (*p* = 0.55) and boundary separation value (*p* = 0.92) were not (Figure 4). Mean non-decision time was associated with NEVHI score but this was not significant after correction for multiple comparisons (*p* = 0.06). Drift rate (*p* = 0.32) and boundary separation value (*p* = 0.45) did not show a significant relationship with NEVHI score. Importantly, UPDRS total motor impairment score did not significantly correlate with non-decision time (*p* = 0.23). This indicates that non-decision time is more strongly related to perceptual measures, rather than measures of motor impairments in this instance.

### 3.6. Clinical Measure of Visual Hallucinations Is Related to Non-Decision Time

To further investigate the relationship between visual hallucinations (VH) and drift diffusion parameters, group comparisons between those patients with a higher NEVHI score (NEVHI+), and those with a low NEVHI score (NEVHI−) were carried out. There was a significant difference in non-decision time between the NEVHI+ and NEVHI− group (U = 183, *p* = 0.024). Boundary separation (*p* = 0.416) and drift rate (*p* = 0.476) did not differ between groups. Finally, Spearman’s Rho correlation showed a significant positive relationship between NEVHI score and non-decision time in MCI-LB patients in the NEVHI+ (rho = 5.412, *p* = 0.013) (Figure 5). However, as the NEVHI score is not a direct indicator of the presence of visual hallucinations, we also carried out a comparison between those patients with the presence of complex VH (VH+), as rated by a clinical panel—as per Donaghy et al. [24]—in accordance with 2017 DLB criteria [41], and patients with absence of complex VH (VH−). VH+ and VH− patients did not show any significant difference in non-decision time (*p* = 0.625) or drift rate (*p* = 0.659), but VH+ patients had significantly higher boundary separation values (U = 90, *p* = 0.002).

## 4. Discussion

Our results showed that mean RT were longer in the Lewy body groups than in the controls, however RT did not differ between possible and probable Lewy body groups, though group numbers were modest. Moreover, accuracy was lower in both Lewy body groups in comparison to controls. These findings are consistent with previous reports of lengthened RTs and reduced accuracy in CRT for Lewy body patients [41,42]. Moreover, previously reported DDM profiles of healthy older adults were characterized by higher boundary separation values with preserved drift rate and non-decision times relative to younger adults. Our findings are consistent with previous findings in that drift rate, and therefore information processing ability, was unaffected by age, while older adults appear to wait to collate more information about the stimuli to ensure an accurate response [17]. This pattern reflects a more cautious strategy to decision making.

In addition, we found that MCI-LB patients had longer non-decision times and lower boundary separation values than healthy older controls but did not differ with regards to drift rate. We propose that this pattern of results suggests that patients had difficulties at perceptual encoding (non-decision) stages of decision making. This suggests that, rather than having impaired overall processing, Lewy body patients have more specific processing deficits in non-decision time, leading to longer RT. Moreover, patients also experienced shorter boundary separation values, suggesting that they adopted a less cautious approach than healthy older adults in their decision making in response to impaired perceptual encoding. This in turn may contribute to lower accuracy. However, no differences were observed in mean drift rates, although it is worthwhile mentioning that patients as a group exhibited increased variability in all DDM parameters. Thus, despite the null effect for mean drift rates, MCI-LB patients drift rate values were more heterogenous, suggesting that perceptual encoding and shorter boundary separation values also affected the rate of information accumulation and hence the quality of the decision-making process. Previous findings in hallucinating PD patients have reported lower drift rates, which indicated less effective sensory accumulation of evidence [19]. Here we did not find any differences in mean but in the variance of the drift rate which suggests that MCI-LB was associated with a greater variability of evidence accumulation. This may mean that MCI-LB patients experienced fluctuations in their ability to effectively process information, in part as a result of perceptual encoding problems, or in part as a result of impairments in alerting and focusing attention. Greater trial-to-trial variability in drift rates would result in slower RT and reduced accuracy, therefore investigating this further may be valuable in future research. As we employed a simple drift diffusion model due to limited trial numbers, this is difficult to recover, therefore future investigations should employ more complex models to examine the relationship between trial-by-trial variability in drift rates and fluctuation in attention. In addition, correlations between DDM variance and cognitive fluctuations may also be of interest, given the transient and variable nature of cognition in MCI-LB and DLB. Moreover, this may further illustrate the variability of cognitive impairment, and its relationship with subsequent clinical presentation in mild cognitive impairment stages of DLB [7].

Approximately 80% of DLB patients experience visual hallucinations [23] that are known to be related to impairments in visuospatial performance (Hamilton et al., 2012). Consistent with a previous study in PD [19] we observed differences between MCI-LB and control groups in non-decision time, a parameter that reflects both perceptual encoding and motor aspects. However, in contrast to the MCI-LB patients in the present study, PD patients had shorter, not longer, non-decision times in comparison to older adults [19]. As non-decision time is comprised of both perceptual and motor aspects, the authors proposed that shorter non-decision times may be partly due to impulsivity-related reductions in response latencies in PD. In the present study, non-decision times were not related to motor impairments, as measured by UPDRS score, but did relate to ACE visuospatial scores, suggesting that differences in MCI-LB patients were most likely due to impaired perceptual encoding. Together, these results suggest that, while the precise mechanisms may differ, both PD and MCI-LB patients experience perceptual encoding impairment that make sensory evidence less informative to them.

The results from our hierarchical stepwise regression fitting revealed that mean non-decision values predicted visuo-spatial performance but not NEVHI score. However, further analyses showed that patients in the NEVHI+ had longer non-decision times than NEVHI− and non-decision time correlated positively with higher NEVHI score in this group. As a positive NEVHI score does not necessarily reflect the incidence of visual hallucinations but can be due to other visual problems [43], we also classified patients on the basis of clinical ratings into those with a history of visual hallucinations and those without. This comparison revealed that individuals with visual hallucinations showed wider boundary separation than those without. Our finding of significantly longer non-decision times in the NEVHI+ versus the NEVHI− group suggests that patients who experience visual difficulties including visual hallucinations are prone to difficulties in perceptual encoding and that these encoding problems may contribute to their experience of visual hallucinations. The additional finding that individuals with a history of visual hallucinations showed larger boundary separation than those without, suggests that these individuals experience additional decision-making problems and adopt a more cautious response strategy in light of noisy perceptual input, that may contribute to and/or reflect the experience of visual hallucinations. Thus, perceptual encoding difficulties appear to be a necessary but not a sufficient condition for the occurrence of visual hallucinations. Such an interpretation is consistent with current theories of visual hallucinations in DLB [21,22] proposing that hallucinations occur as a product of impaired perceptual encoding or bottom-up processing, and subsequent over activation of top down processes, which may result in incorrect objects being inserted into the visual scene, i.e., visual hallucinations. In this context, our results may demonstrate that the processing of sensory stimuli is slower and potentially less informative in all MCI-LB patients but that in patients who hallucinate such impaired sensory processing is accompanied by additional top-down decision-making problems.

In contrast to our findings, O’Callaghan et al. [19] reported that all PD patients showed wider decision boundaries, but only hallucinating patients showed slower drift rates than those patients who did not hallucinate. While this pattern of results also supports the view that impaired sensory evidence accumulation may lead to reduced information processing quality and an over-reliance on top down processing in PD, which in turn may underpin visual hallucinations, the mechanisms leading to these impairments seem to differ between PD and MCI-LB.

There were limitations of the present study. Due to practicalities in recruitment, healthy control participants were significantly younger on average than probable and possible Lewy body patients, which may have affected some drift diffusion estimates. However, age was not a significant predictor in the linear regression analyses. In addition, due to recruitment from a University panel, control participants had significantly more years of education than Lewy body groups, which may also have influenced performance in some cognitive tests. We assessed the potential influence of age and education on our results by comparing patients with a subgroup of controls that were closely matched for these variables. These additional analyses resulted in the same pattern of results, suggesting that the group differences in age and education did not drive our findings.

Moreover, although tasks were similar in nature between healthy control and patient groups, there were some notable differences in both trial number and stimuli which could influence RT and accuracy and bias the DDM modelling. Firstly, trial numbers were relatively low in the CRT in MCI-LB groups, due to practical limitations and patients’ task completion ability, which may influence the accuracy of model fitting. Secondly, there is possibility that the presence of neutral flankers in the healthy control task may produce lower drift rates due to crowding effects, therefore resulting in an under estimation of the current group differences in our results. Thirdly, as control participants also took part in randomly presented congruent and incongruent trials, boundary separation may be raised as older adults become more cautious in response overall, thus overestimating the boundary separation effect in this study [44]. Finally, non-decision time may be influenced by a more complex stimuli, resulting in an under-estimation of the observed perceptual encoding group differences in the present study. This does pose some limitation to the conclusions which can be drawn, therefore replication in a larger sample, and with identical task conditions is vital to clarify these effects. Despite this, employing the same task between patients and control subjects is also challenging, as those tasks which are accessible for patients in terms of trial number or stimuli may be very simple for controls to complete, resulting in a ceiling effect, and vice versa.

To conclude, the present findings suggest that impaired perceptual encoding, as estimated by the DDM, contributes to NEVHI score in MCI-LB, and altered boundary separation may be related to the presence of visual hallucinations [19,45].

## Figures and Tables

**Figure 1 brainsci-10-00540-f001:**
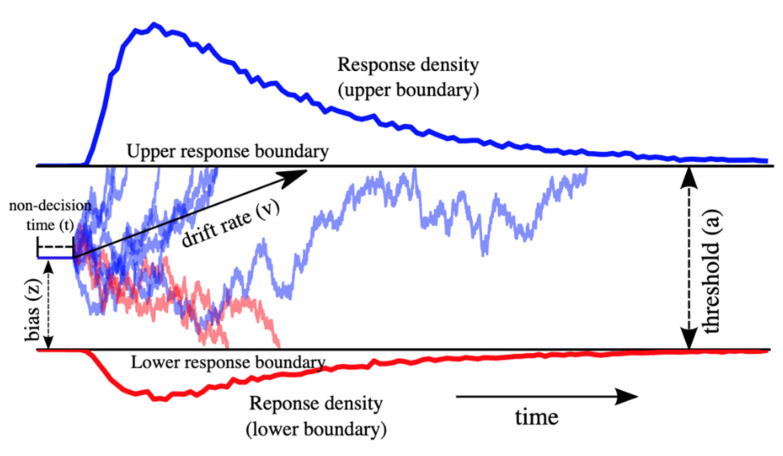
The drift diffusion model. Red and blue denote the information being accumulated during one trial towards one of two responses in a 2-choice task (for example, left or right). Lines in the center of the figure represent the noisy accumulation of evidence for either decision during a trial. In this instance, more evidence is accumulated toward a blue (upper) response. Reaction times from each trial are fit to this model to return an average estimate of non-decision time (t), boundary separation or threshold (a), and drift rate (v). Adapted from Wiecki et al. [16].

**Figure 2 brainsci-10-00540-f002:**
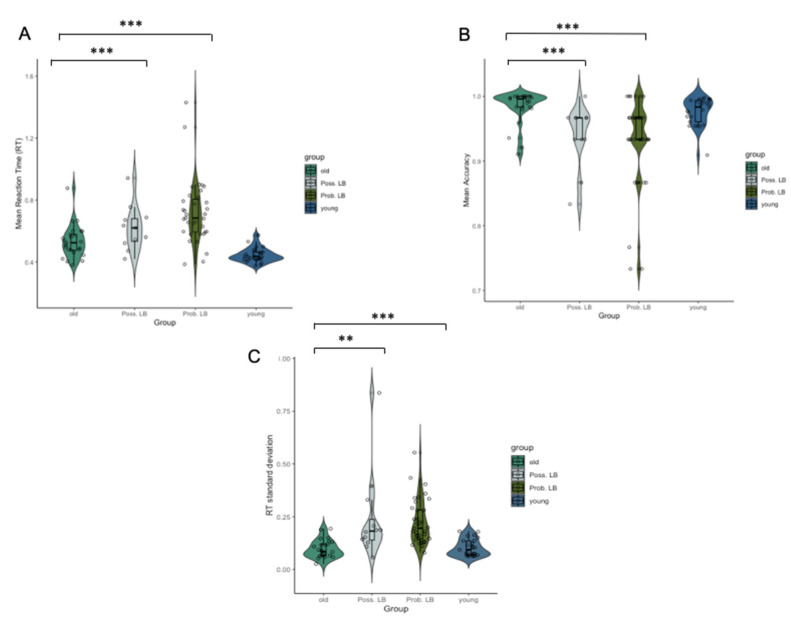
(**A**) Diagnostic group differences in mean reaction time (RT) (**B**) Group differences in mean Old = older healthy controls, young = younger healthy controls, Poss. LB = possible MCI-LB, Prob. LB = probable MCI-LB. (**C**) Standard deviation of RT is significantly greater in possible and probable MCI-LB than older adults. ** *p_cor_* = 0.01, *** *p_cor_* < 0.001.

**Figure 3 brainsci-10-00540-f003:**
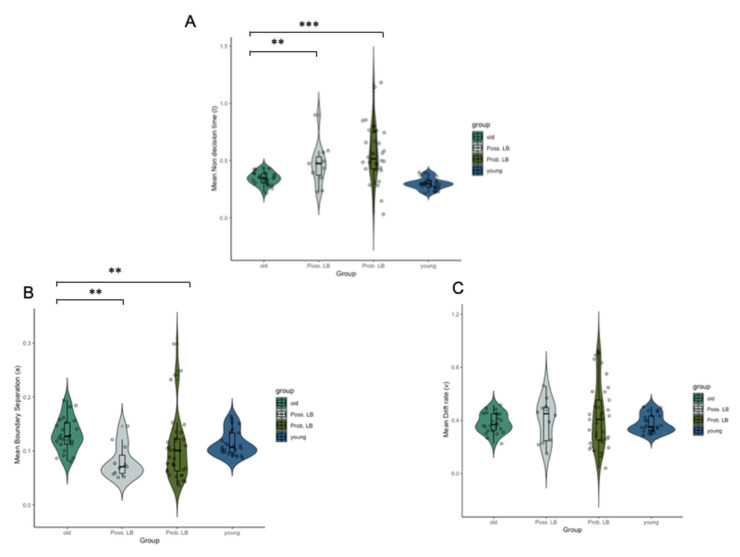
Diagnostic group differences in (**A**) non-decision time (t) and (**B**) boundary separation (a) and (**C**) drift rate DDM values. ** *p_cor_* = 0.01, *** *p_cor_* ≤ 0.001.

**Figure 4 brainsci-10-00540-f004:**
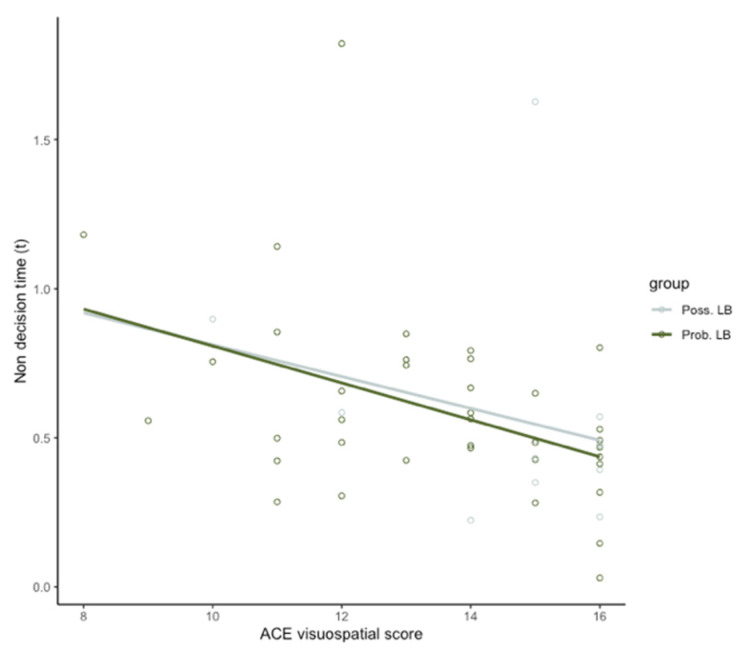
Non-decision time (t) is negatively related to scores on the Addenbrooke’s Cognitive Exam (ACE) Visuospatial subscale in mild cognitive impairment (MCI-LB) groups (adj *R*^2^ = 0.129, beta = −0.333, *p* = 0.014).

**Figure 5 brainsci-10-00540-f005:**
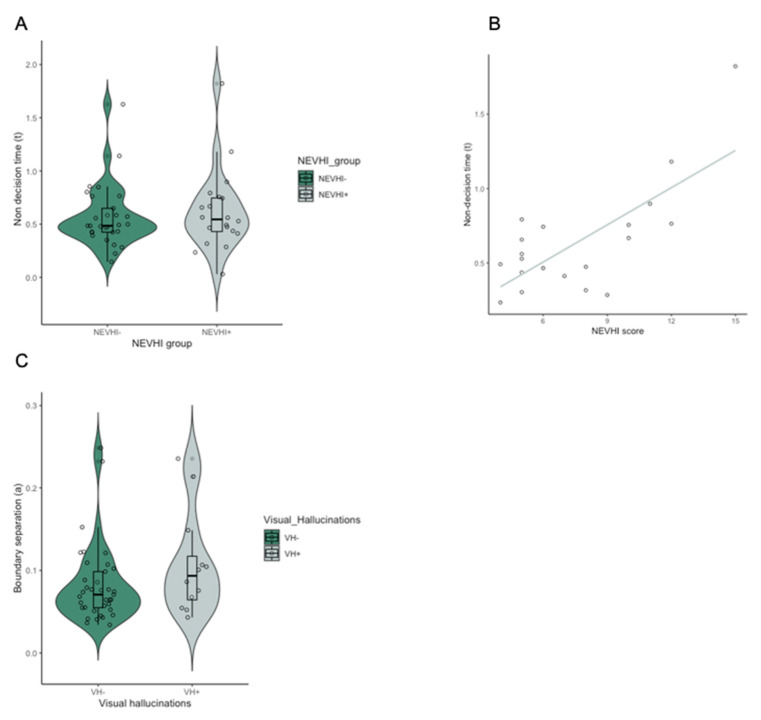
(**A**) Non-decision time (t) comparisons between NEVHI+ and NEVHI− MCI-LB patients (U = 183, *p* = 0.024). (**B**) Relationship between visual hallucinations score (NEVHI) and non-decision time in VH group (rho = 5.412, *p* = 0.013). (**C**) Boundary separation (a) comparisons between VH+ and VH− MCI-LB patients (U = 90, *p* = 0.002).

**Table 1 brainsci-10-00540-t001:** Demographic information of Lewy-Pro cohort and control participants. ACE = Addenbrooke’s Cognitive Examination, NEVHI = North East Visual Hallucinations Inventory, UPDRS = Unified Parkinson’s Disease Rating Scale, CAF = Clinical Assessment of Fluctuations.

	Probable MCI- LB (*n* = 37)	Possible MCI- LB (*n* = 12)	Older Controls (*n* = 25)	Younger Controls (*n* = 25)
Age	72.86 (15.51)	75.25 (7.3)	68.36 (6.11)	21.69 (2.85)
Sex	Female = 13	Female = 3	Female = 13	Female = 15
Education	11.57 (2.85)	10.75 (2.09)	15.12 (2.40)	14.95 (1.89)
ACE	78.34 (8.88)	79.33 (14.09)	93.52 (4.18)	92.38 (3.12)
Visual Hallucinations	20	2	-	-
NEVHI	3.66 (4.47)	1 (3.316)	0	0
UPDRS	25.21 (15.71)	15.36 (7.75)	-	-
CAF	2.70 (3.03)	2.16 (2.48)	0.32 (0.63)	0
Cholinesterase Inhibitor	18	0	-	-
Levodopa	8	0	-	-

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
