# Peer review of "Visuo-Perceptual and Decision-Making Contributions to Visual Hallucinations in Mild Cognitive Impairment in Lewy Body Disease: Insights from a Drift Diffusion Analysis"

_brainsci, 2020, doi:10.3390/brainsci10080540_

Round 1

Reviewer 1 Report

This manuscript reports on an observational study of the cognitive mechanisms of visual hallucinations in Lewy bodies dementia. This attempt is original and the results of this study deserve to be published.

The sample is appropriate (diagnosis of LB-MCI, size). The model used by the authors is very complex and difficult to understand. In particular, Figure 1 needs further explanation. It seems to have several y-axes, but they are not defined. Do the blue and red lines in the centre of the figure represent measures of reaction time? If so, why are there more blue than red responses? Why did the reaction time measurement stop before the end of the experiment (response density)?

There are no significant differences between patients with possible LB and probable LB MCI, in terms of their characteristics and outcomes. The rationale to make separate groups is unclear. It may be simpler to make a single group of LB-MCI and to indicate in a few words in the text the comparison between the two subgroups of LB-MCI.

About the analysis of visual hallucinations: it is not clear what a high or low NEVHI means and how many patients had a high or low NEVHI. The number of patients with and without visual hallucinations according to the clinical panel should also be indicated, for example in Table 1.

Reviewer 2 Report

Overall great paper, original concept.

Several grammatical corrections need to occur throughout the paper. Numerous run-on sentences that need to be divided into 2 or more sentences, few examples are listed below, however recommend reducing the use of the word "which" as it is used very frequently throughout the manuscript. Recommend avoiding use of passive language in the paper. Use acronyms consistently, ie: can use "DDM" throughout vs going back and forth, previously used acronyms do not need to be reintroduced ie: line 389 "RT" this was noted previously. 

Line 59: "adopted the diffusion drift model.." Add the "(DDM) here as this is where it is first introduced, and remove from line 62. 

Line 62: The drift diffusion model [14,15] is a sequential sampling model, which assumes that information which drives a decision is accumulated over time, until it reaches one of two response boundaries, which form the ultimate response (for example, ‘left’ 64 or ‘right’ would be response boundaries in one model; Figure 1). Recommend grammatical corrections to this sentence, too many "which" run-on sentence.

Line 67-70:Recommend grammatical corrections to this sentence, too many "which".For example: "according to the diffusion model, overall processing is segmented into several components contributing to ultimate performance: 

Consider numbering each component such as (1) ... , (2).., and (3)...  along with grammatical corrections. 

Line 71: Remove DDM - this should be listed the first time the words drift diffusion model are introduced into the manuscript. 

Figure 1: caption "adapted from ** List author et al. (16)

Line 94: Replace to say Parkinson Disease (PD) patients; "relative to those without" clarify as this study compared PD patients with hallucinations to those without and controls. Grammer- "in those with hallucinations"

Line 95:"Slower drift rates were found in those who hallucinated (change to PD patients with hallucinations), and shorter perceptual encoding times were found in PD patients (are these patients with hallucinations or without?) compared to controls, suggesting that the accumulation of evidence was  hindered by perceptual encoding problems" 

Line 109: caregivers one word

116: Example on how to reword this to improve grammar and flow. Study participants were part of the Lewy-Pro cohort previously described in detail [24,25] Briefly, individuals were recruited from memory clinics in the.., were greater than 60 years of age, and met.. for MCI. All participants were clinically assessed for core features of DLB by an experienced psychiatrist. Additional assessments included neuropsychological testing and  dopaminergic imaging testing.. Also clarification on the cohort, individuals were "offered" a FP-CIT SPECT - do you mean to say they all received it or "all individuals underwent a FP-CIT SPECT scan? 

123: Old age is not a commonly used term in all forms of English, if this is to describe an experienced physician I would change it to experienced vs a fellowship trained, etc. 

130: conditions

Table 1 corrections - Caption "Lewy-Pro", Addenbrooke’s Cognitive Examination (it is not plural), Unified Parkinson’s Disease Rating Scale, Clinical Assessment of Fluctuations (and spacing).Why was there a CAF completed on the older controls? Recommend reporting the ranges along with std dev. in the table.

158: completed a simple 30 trial - do you mean to say 30 trials?

180:"although also see [39]" unclear what this means.

Statistical analysis: were individuals controlled for duration of MCI? 

Figure 2: caption grammar corrections, and define old and young, recommend "older controls" and "younger controls" (congruent with what is written in table 1) Figure 2b - missing part of the bracket line. Capitalize words consistently.

Post-hoc, vs post hoc - be consistent throughout. correct spacing in between = signs etc. Make font size on legends larger. 

Section 3.4. The potential influence of age and education on group differences was assessed given that the participants with possible and probable MCI-LB were older and had less educated compared to the older control group. 

Lines 313 - 320 spacing is off, paragraph indentation. and non-decision time (be consistent with dashes)

Line 346 - why VH here?

354- as per..list author et al with ref.

Figure 5 - legend remove _ 

line 406 suggesting they adopted a less

Line 415 This may mean that MCI LB patients experienced fluctuations in their ability to effectively process information, in part as a  result of perceptual encoding problems, or in part as a result of impairments in alerting and focusing attention. Were the tests compared to individuals with higher vs lower CAF scores?

Line 471 Recommend limiting use of the phrase "it should be noted" as this was used often.

Discussion does a good job explaining the meaning of the results, however there is some redundancy in repeating the results. 
